# Bayesian network modeling of patterns of antibiotic cross-resistance by bacterial sample source

Stacey S. Cherny [1,2], Michal Chowers[3,4] & Uri Obolski [1,2✉]

## Abstract

**Background** Antimicrobial resistance is a major healthcare burden, aggravated when it extends to multiple drugs. While cross-resistance is well-studied experimentally, it is not the case in clinical settings, and especially not while considering confounding. Here, we estimated patterns of cross-resistance from clinical samples, while controlling for multiple clinical confounders and stratifying by sample sources.

**Methods** We employed additive Bayesian network (ABN) modelling to examine antibiotic cross-resistance in five major bacterial species, obtained from different sources (urine, wound, blood, and sputum) in a clinical setting, collected in a large hospital in Israel over a 4-year period. Overall, the number of samples available were 3525 for *E coli*, 1125 for *K pneumoniae*, 1828 for *P aeruginosa*, 701 for *P mirabilis*, and 835 for *S aureus*.

**Results** Patterns of cross-resistance differ across sample sources. All identified links between resistance to different antibiotics are positive. However, in 15 of 18 instances, the magnitudes of the links are significantly different between sources. For example, *E coli* exhibits adjusted odds ratios of gentamicin-ofloxacin cross-resistance ranging from 3.0 (95% CI [2.3,4.0]) in urine samples to 11.0 (95%CI [5.2,26.1]) in blood samples. Furthermore, we found that for *P mirabilis*, the magnitude of cross-resistance among linked antibiotics is higher in urine than in wound samples, whereas the opposite is true for *K pneumoniae* and *P aeruginosa*.

**Conclusions** Our results highlight the importance of considering sample sources when assessing likelihood of antibiotic cross-resistance. The information and methods described in our study can refine future estimation of cross-resistance patterns and facilitate determination of antibiotic treatment regimens.

## Plain language summary

Antibiotics are drugs that kill some bacteria. Antibiotic resistant bacteria are bacteria that continue to grow despite the presence of an antibiotic drug. These bacteria are a major problem in healthcare, particularly if the bacteria are resistant to multiple drugs. Here, we study bacteria that are resistant to several antibiotics that are present in patients in hospital. We find that patterns of cross-resistance differ between the location bacteria were sampled from, such as blood or urine. Our results highlight the importance of considering sample sources when assessing the likelihood that bacteria is resistant to multiple antibiotics. The information and methods described in our study should enable further analysis and prediction of the presence of cross-resistant bacteria, enabling appropriate antibiotic treatments to be used.

[1] School of Public Health, Tel Aviv University, Tel Aviv, Israel. [2] Porter School of the Environment and Earth Sciences, Tel Aviv University, Tel Aviv, Israel. [3] Meir Medical Center, Kfar Saba, Israel. [4] Sackler School of Medicine, Tel Aviv University, Tel Aviv, Israel. ✉email: uriobols@tauex.tau.ac.il

Antimicrobial resistance is a substantial healthcare burden, causing increased use of hospital resources, changes in treatment protocols, and excess morbidity and mortality[1]. When bacteria are resistant to multiple drugs, the problem is exacerbated. As a result, studying bacterial resistance is a major area of inquiry[2]. Many predictors of resistance in clinical settings have been identified, including age, patients' independence status, and previous antibiotic usage[3,4]. Additionally, the interrelationship of resistance between different antibiotics is of primary clinical relevance.

Cross-resistance is the phenomenon where a bacterial isolate that is susceptible (or resistant) to a particular drug, will often be susceptible (or resistant) to a different drug. Importantly, in this study we will only refer to cross-resistance due to mechanisms that are observed to be acquired through genetic changes, and not those inherent, for example, to a certain bacterial species. Cross-resistance of bacteria to different drugs is a commonly observed phenomenon in experimental[5–12] and clinical settings[13–17]. It is especially prevalent between antibiotics from the same antibiotic class, although differences in the drugs may lead to imperfect associations[12]. On the other hand, discordant resistance is the phenomenon where susceptibility to one drug is associated with resistance to another (and vice versa). Discordant resistance is often demonstrated in experimental studies[5–10]. The experimental paradigm typically involves exposure of bacterial cultures to increased doses of a single antibiotic over many generations, and then testing the adapted population for resistance to other antibiotics. However, such experimental conditions do not necessarily mimic bacteria living in human populations. As a result, it is perhaps not surprising that discordant resistance is rarely observed in clinical settings[13,14]. Nonetheless, a recent paper proposed a method of analysis of clinical MIC data that is somewhat analogous to the experimental approach, that not only examines the existence of cross or discordant resistance, but attempts to establish direction of effect[14]. They found some of the limited discordant (and cross) resistance present was unidirectional.

Revealing cross-resistance patterns in clinical settings is not trivial. Observational data are often subject to confounding due to non-random exposure to antibiotics, which can substantially mask the underlying cross-resistance patterns. Additive Bayesian network (ABN) modeling is a data-driven approach to inferring underlying probabilistic structure of a set of variables. By essentially searching all possible directed acyclic graphs (DAGs) linking a set of variables, direct vs indirect connections between variables can be uncovered from the data. ABNs perform this task without strong prior assumptions – a task which is difficult to achieve using standard regression methods. Such an approach is useful when resistance to many different drugs and the presence of several important covariates need to be analyzed simultaneously, but strong assumptions regarding many of the connections are not available. To our knowledge, we are the only ones to have previously employed ABN modeling to deal with such confounding in antibiotic resistance in clinical data[15]. Nonetheless, the method has previously been used for examining resistance networks in veterinary data[18], and in clinical data but without controlling for covariates[19]. In addition, a related network modeling approach, chain graphs, has also been recently used to examine antibiotic resistance networks in clinical data[20]. ABN models allow uncovering the association structure among a set of variables, while controlling for relevant confounders. However, the differences of cross-resistance patterns between sample sources have not been explored in depth previously, while adjusting for confounders. Bacteria from different sample sources may have undergone divergent evolutionary trajectories as a result of different selective forces, perhaps leading to distinct patterns of resistance. Such differences can have clinical implications on short- and long-term antibiotic treatment selection.

In this study, we use a large clinical dataset obtained from an Israeli hospital over a 4-year period, to study cross-resistance in different sample sources. We employ ABN modeling to control for potential confounding variables and explore the resistance network structures by bacterial species and sources of the bacterial sample. This allows for unbiased examination of the patterns of cross-resistance, whether they differ by sample source, and which variables can predict those patterns of resistance. We found that the magnitude of cross-resistance substantially differs across sample sources, highlighting the importance of considering sample sources when assessing likelihood of antibiotic cross-resistance.

## Material and methods

**Data**. We obtained data pertaining to all positive bacterial cultures drawn in Meir Medical Center, Israel, a 740-bed hospital with ~60,000 admissions per year, from 2016-01-02 to 2019-12-31. The corresponding medical history, demographics, previous hospitalizations, and previous in-hospital antibiotic usage in the year prior to the infection, of patients from whom the samples were drawn, were also available. Bacterial samples were tested for antibiotic resistance for an array of antibiotics, and results of non-susceptibility and resistance were combined into a 'resistant' category. All antibiotics tested were systemic, with the exception of mupirocin, which is topical. Nonetheless, given we had data on bacterial resistance to mupirocin, it was still included in the analyses. Bacterial infections were considered nosocomial if samples were drawn >48 h after admission. A summary of these variables is presented in Supplementary Data 1. For our analyses, we selected the five bacterial species with the largest sample sizes available in the dataset: *Escherichia coli*, *Staphylococcus aureus*, *Pseudomonas aeruginosa*, *Klebsiella pneumoniae*, and *Proteus mirabilis*, in order of decreasing frequency.

**Statistics and reproducibility**. We selected which antibiotics to include in the analysis by keeping only those with minimal missing data and which did not reduce the number of complete cases appreciably (<10% loss). We performed some variable selection to assure stable statistical models with no perfect or near-perfect separation, by not including perfectly or near perfectly correlated antibiotics and selecting only antibiotics which contained a minimum of 3% resistance in each bacterial subsample. Antibiotics excluded from analysis due to high collinearity with included variables are presented in Table S1, along with their range of tetrachoric correlations with the relevant included variables, across the various sample sources. Excluded antibiotics were usually, but not always, of the same drug family. For example, while ceftazidime, ceftriaxone, cefuroxime, and cefalexin are all cephalosporins and we often removed three of these four, ampicillin, a beta-lactam, was removed due to high correlation with the cephalosporins. This resulted in analysis of between three and five antibiotics for the five bacterial species cultured from the various sources, each analyzed separately.

When constructing the ABN, the following covariates were included, in addition to the antibiotic resistance tests: demographic variables (age, sex, and days hospitalized in the previous year), presence of five medical conditions (immunosuppression, dementia, diabetes, chronic obstructive pulmonary disorder (COPD), chronic renal failure (CRF), and obesity (BMI over 30), binary sample type variables (nosocomial and polymicrobial), and a binary variable for antibiotic use in hospital in the previous year. Although modeling previous antibiotic use is preferable as continuous with an interaction of a binary

used/didn't use term[21], this could not be achieved under the ABN framework, so we resorted to the mentioned binarization, as we have successfully done previously[15].

Antibiotics used were first grouped into drug families and the six most frequent families across the entire dataset were included, along with whether any other antibiotics were taken (which did not belong to the six largest families). The six families were (in descending frequency of use) cephalosporins, beta-lactams (that are not cephalosporins), beta-lactamase inhibitors, nitroimidazoles, aminoglycosides, and fluoroquinolones, plus a seventh category including all other antibiotics.

Supplementary Data 1 presents summary statistics for all variables used in our models, separately by bacteria. For some models, certain covariates needed to be excluded from the analysis in order for the ABN models to converge due to insufficient sample size. In addition, for the *E coli* urine analysis, to allow the model to run within a reasonable amount of time (days rather than months), two variables were omitted which did not have a direct connection with resistance in the initial search, as noted in the spreadsheet.

Data were analyzed using ABN modeling[22,23], with version 2.5.0 of the R package abn[24] on an R 4.1.0 installation[25], using a procedure previously employed by us[15]. While no assumptions of causal relationships are required for our analysis, we restricted the model space to disallow causal paths that were not clinically plausible: (1) sex could only cause other variables and not be caused by any other variable; (2) age at testing could be caused by *sex* but no other variable; (3) days hospitalized the prior year could not be caused by presence of a nosocomial infection or any resistance test, but could be caused by any other variable; (4) nosocomial infection could not be caused by any resistance test but can be caused by any other variable; (5) polymicrobial cultures as well as all antibiotic resistance tests could be caused by any of the variables; (6) the variables for antibiotics given in the previous year could only be caused by age and sex and no other variable; (7) immunosuppression and CRF can be caused only by any of the other medical conditions as well as age and sex; (8) obesity can only be caused by dementia, age, and sex; and (9) dementia, diabetes, and COPD can be caused only by each other or obesity (but not immunosuppression nor CRF), as well as age and sex. In addition, no causal paths were forced to be in the model. These restrictions correspond to a matrix of banned arcs, presented in Fig. S1.

We followed a multi-step procedure to fit the ABNs: first identifying the most-likely network structure, using the abn package's exact order based structure discovery approach, with an uninformative prior, scoring networks based on their Bayesian information criterion (BIC), to find the most probable posterior network[26]; then performing a parametric bootstrap, using JAGS[27] to simulate 1000 datasets per model; then re-fitting ABNs to reanalyze each simulated dataset; and finally, re-estimating parameters using ABNs, only allowing for connections seen in at least 50% of the analyses of the simulated datasets, to correct for overfitting, and computing 95% Bayesian credible intervals (CIs) for each of the parameter estimates. The procedure we employed has been previously described and applied to antibiotic

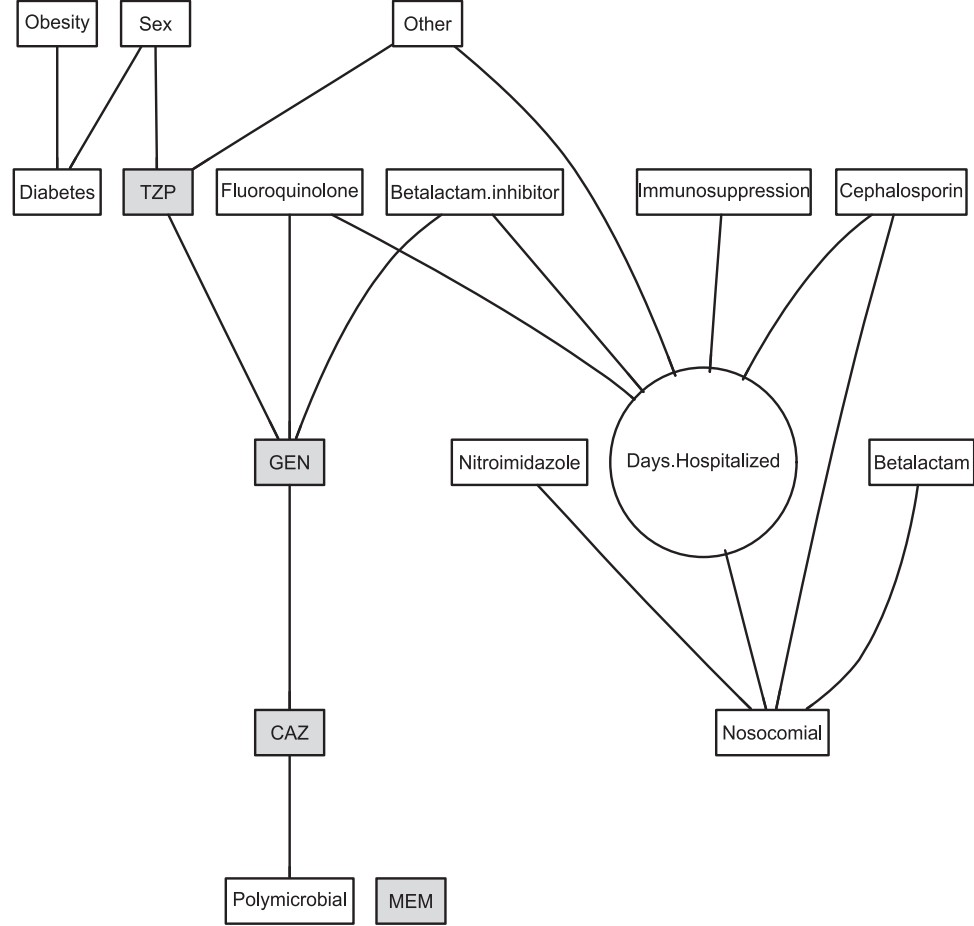

**Fig. 1 Additive Bayesian network for *P aeruginosa* in wound samples.** Variables in rectangles are binary and the one in a circle is continuous. Drug family variables refer to a particular drug taken in the prior year. Antibiotic resistance tests are shown in shaded boxes: CAZ ceftazidime, GEN gentamicin, MEM meropenem, TZP piperacillin/tazobactam.

resistance data by us[15]. However, to speed up the parametric bootstrap in the present study, we dropped all variables that had no connection to any other variable in the initial network and reran the full procedure on the reduced dataset. This would have no effect on the results presented here. In addition, we limited the search for the maximum number of parents to eight, due to prohibitive computational time needed to conduct the search allowing more parents. Table S2 contains the number of arcs present in the model both before and after bootstrapping, with 7 parents being the most any model required. Hence, this restriction did not affect our results. While the ABN methodology generates results with directional arcs, due to potential identifiability issues stemming from different structures having the same likelihood values, we present results with direction omitted, as suggested in previous studies[22,28].

To test whether arcs present in different sources of the same bacterial species were significantly different in magnitude, we randomly sampled from the posterior distributions of a given arc and computed the difference between a random pair posterior ln(OR) estimates, which yielded a distribution of ln(OR) differences. From that, we present the 95% CIs of the differences and examine whether zero was inside the intervals, indicating no significant difference in arc parameter estimates. The posterior estimates were computed at a granularity of 10,000 per parameter and 100,000 pairs were sampled with replacement to generate the empirical distribution of ln(OR) differences.

**Ethics**. The study was approved by the Institutional Review Board (Helsinki) Committee of Meir Medical Center. Since this was a retrospective study, using archived medical records, an exemption from informed consent was granted by the Helsinki Committee.

## Results
In Fig. 1, we present an example of an ABN estimated for *P aeruginosa* in a wound sample. The antibiotic resistance tests had direct links to sex, polymicrobial infection, and previous use of fluoroquinolones, beta-lactam inhibitors, and less-common antibiotics (other). However, resistance to MEM was not linked to any variable. Thus, the network yields less biased associations between the different antibiotics by controlling for various patient covariates. ABNs for all bacterial species investigated are presented in Figs. S2–15, with parameter estimates and 95% CIs presented in Supplementary Data 2–15.

To better summarize our results, we present consensus graphs (Fig. 2). These graphs present the relationships among drug resistances from all bacterial sources for a given bacterial species. Figure 2 contains 5 panels, a-e, representing the relationships among the antibiotic resistance tests in *E coli, K pneumoniae, P aeruginosa, P mirabilis, and S aureus*, respectively, obtained from fitting the ABN models (as the one presented in Fig. 1). The nodes in the consensus graphs are denoted with both the antibiotic resistance tests included in the ABN models and corresponding antibiotics excluded due to very high correlations with the included antibiotic, as seen from examination of the pairwise tetrachoric correlations (Table S1).

In Fig. 2, arcs are color-coded by sample source: orange arcs are in urine, red in wound, blue in aerobic blood, and black for arcs present in all three sources for a particular bacterial species. In all four of the species where ciprofloxacin and gentamicin were tested, they were linked, with the exception of *P mirabilis*, where they were linked in two of the three sample sources. Ceftazidime and gentamicin were also linked in four species: in *P aeruginosa*, in all three sources and in two of three sources in *E coli* and *P mirabilis*. Piperacillin/tazobactam was directly linked to ciprofloxacin in two of the three species for which it was tested and in

two sample sources of *K pneumoniae*. Importantly, all direct connections were consistently positive throughout all antibiotics and all bacterial species.

Finally, we sought to analyze whether the magnitude of the cross-resistance links differed between sample sources. Parameter estimates of the cross-resistance links, along with their 95% CIs, are presented in Fig. 3 (with the data used to create this figure presented in Supplementary Data 16) for those parameters estimated in more than one sample source of a particular bacterial species. The median differences between these parameter estimates in different sample sources, along with 95% CIs for the difference, within bacterial species, are presented in Table S3. While many arcs were present in more than one sample source for a given species, only in three of 18 instances were the magnitudes of those parameter estimates not significantly different between sources compared.

In *E coli*, the link between ofloxacin and ceftazidime was not significantly different in aerobic blood than in urine, as was the link between ceftazidime and gentamicin in urine versus wound. The third pair of estimates not significantly different was cefuroxime and ofloxacin in the *P mirabilis* samples from urine versus wound. All other links present in two different sample sources were found significantly different between sources, as determined by the 95% CIs of the differences not including zero. Regarding the magnitude of cross-resistance, for *P mirabilis*, all significant differences resulted in higher cross-resistance for urine than for wound samples. Conversely, for *K pneumoniae* and *P aeruginosa*, cross-resistance was higher in wound than in urine samples. *E coli* presented no such clear patterns.

## Discussion
Bacterial cross-resistance has been studied in experimental[5–11] and clinical settings[13–17]. However, to the best of our knowledge, cross-resistance patterns have not been compared between clinical sample sources, while controlling for biases arising from the retrospective nature of such data. In this work, we examined patterns of cross-resistance across bacterial sample sources, while adjusting for potential confounders, using ABNs. We found that the patterns and magnitude of cross-resistance among pairs of antibiotics may vary significantly between sample sources.

The general patterns of cross-resistance in our study are consistent with previous research. For example, all identified links were positive; and links within antibiotic classes, with similar resistance mechanisms, were high. In fact, the cross-resistance estimates of some antibiotics could not be disentangled in our analysis, due to their high correlations. Such a grouping due to high correlations was formed by the cephalosporins ceftazidime, ceftriaxone, cefuroxime, and cefalexin in *E coli, K pneumoniae, and P mirabilis*, for all tested sample sources. As expected, resistance to older generations of cephalosporins did not necessitate resistance to newer generations, but almost never vice versa. However, because those instances were relatively rare, the correlations between antibiotics remained very high. Some discrepancies were also found between our study and a large and recent study of cross-resistance in clinical bacterial samples[13]. The differences between the results could be due to several major differences between the analyses. Firstly, we controlled for an array of relevant covariates using ABNs. Secondly, we examined relationships among resistances stratified by sample sources. Finally, systematic differences in healthcare systems, antibiotic prescription practices, or cohorts of patients may also contribute to such discrepancies. For example, our data are relatively homogenous, since they originated from a single center. In contrast, the data used in the mentioned study originated from over 35 separate hospitals across the US[13]. In another study[20], a

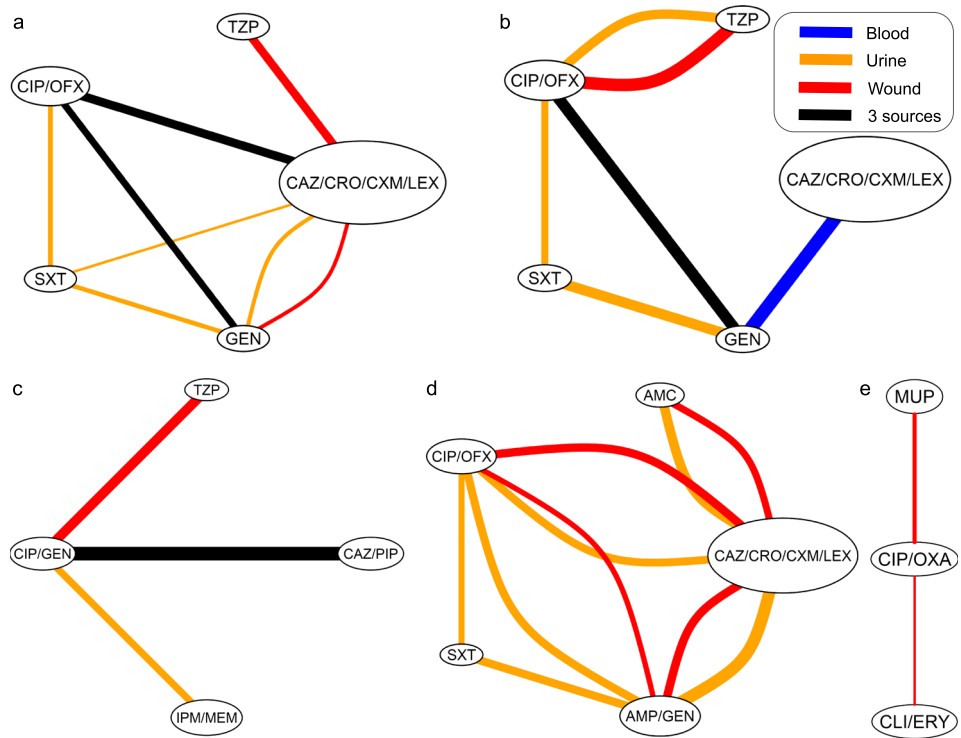

**Fig. 2 Consensus network graphs for the five bacterial species. a** *E coli*, **b** *K pneumoniae*, **c** *P aeruginosa*, **d** *P mirabilis*, and **e** *S aureus*. Line color depicts in which bacterial source model the link is present, with line width proportional to the ln(OR) (or mean ln(OR) for black lines). Orange = urine; red = wound; blue = aerobic blood; black = urine, blood, and wound, except in *P aeruginosa*, where black denotes urine, blood, and sputum. AMC amoxicillin/clavulanate, AMP ampicillin, CAZ ceftazidime, CIP ciprofloxacin, CRO ceftriaxone, CXM cefuroxime, ERY erythromycin, GEN gentamicin, IPM imipenem, LEX cefalexin, MEM meropenem, MUP mupirocin, OFX ofloxacin, OXA oxacillin, PIP piperacillin, TZP piperacillin/tazobactam, SXT sulfamethoxazole/trimethoprim.

somewhat similar approach of chain graphs has been implemented to identify patterns of cross resistance while controlling for clinical covariates. Nevertheless, the study did not allow network structure to vary by sample source, and only included MRSA infections, which are highly resistant by definition. Moreover, the antibiotic resistance results examined in the mentioned study only partially overlap with those studied here. Hence a direct comparison of the results is not possible.

The present study extends on our previous research, applying the flexible ABN methodology to identification of antibiotic resistance network structure[15]. Although cross-resistance has been studied epidemiologically[13,15,20,29], such studies rarely compare cross-resistance patterns of different bacterial sample sources[13]. To the best of our knowledge, ours is the first study performing such a comparison while controlling for relevant clinical covariates. When doing so, all cross-resistance links identified in our study were positive or non-existent, and 15 of 18 comparable pairs of links were significantly different in their magnitude between different sample sources. This suggests the broad conclusion that the extent of cross resistance found might depend on the selective pressures exerted by the environment the bacteria were sampled from.

From a practical standpoint, our study offers two main advances. First, highlighting the consideration of source-dependent cross-resistance patterns. Studying cross-resistance in situ is of critical importance, since bacterial conditions in infections may substantially differ from in vitro conditions, leading to different adaptations[30,31]. In our study, we found that in several cases, the differences of cross-resistance magnitude between sample sources were dramatic. For instance, in *K pneumoniae*, the OR between gentamicin and ofloxacin ranged from around 6 in urine to nearly 40 in blood. In *E coli*, the same

antibiotics varied in their ORs from 3 in urine to 11 in blood. These differences can distinguish between a treatment being merely unlikely to succeed, given one type of resistance, to being almost futile. Moreover, despite the complexity of the results, one general trend can be discerned: *P mirabilis* exhibited higher cross-resistance in urine than in wound samples, whereas the opposite was true for *K pneumoniae* and *P aeruginosa*. Such information could be incorporated into local clinical guidelines or decision support systems. For example, using the above estimates, if a certain hospital has high frequencies of ofloxacin resistance, an immediate implication is that gentamicin would be futile to use for bloodstream infections but may still be efficient in urinary tract infections. Second, we provide code and detailed description of the methods used to obtain our results. This will enable researchers and clinicians to analyze covariate-adjusted cross-resistance patterns on their particular datasets, and draw their specific conclusions, which may vary from ours, as discussed below.

There are several limitations to our analyses. First, sample size differed greatly both in bacterial species and sample sources. This translates into differing levels of statistical power across ABN analyses. For example, we observe the most arcs in the network for *E coli* obtained from urine, potentially because the sample size was largest in that analysis. For the three other bacterial species cultured from urine, it is apparent that there are more arcs in the models connecting antibiotic resistance tests as compared to other sample sources, again potentially due to urine having the largest sample sizes. Nonetheless, while sample sizes were modest in some groups, ample data was available for others, and the study examined all bacterial samples obtained in a large hospital over a 4-year period, making it quite comprehensive. Moreover, this limitation only pertains to lack of identification of further

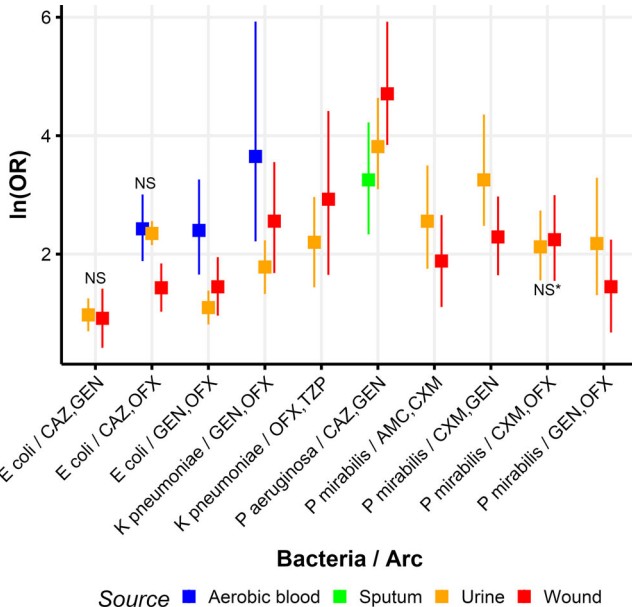

**Fig. 3 Parameter estimates (ln(OR)) and their 95% credible intervals, for arcs between pairs of antibiotics, by bacteria and sample source, where both antibiotics appear in more than one sample source for a given bacterial species.** NS near a pair denotes that the two estimates were not significantly different, with asterisk denoting within 0.01 from the null. The 95% CI of the estimate of the difference between all other pairs tested within a bacterial species did not contain zero. Line color depicts in which bacterial source the estimate was obtained: orange = urine; blue = aerobic blood; red = wound; green = sputum. AMC amoxicillin/clavulanate, CAZ ceftazidime, CIP ciprofloxacin, CXM cefuroxime, GEN gentamicin, OFX ofloxacin, TZP piperacillin/tazobactam. *The 95% credible interval contains zero to one decimal place.

links. When we do present identified links, the influence of the sample sizes is exhibited in our uncertainty quantifications (e.g., CIs), which did successfully demonstrate differences between most sample sources.

Secondly, the patterns found in our analyses are not necessarily generalizable to other countries or even hospitals. Resistance patterns are determined by prescription patterns, patient demographics and a host of other factors, and are expected to differ across time and space[4,32]. To draw exact conclusions for a specific hospital, one would need to fit our models to the respective data, as resistance patterns may differ substantially between hospitals. This would entail retraining the models with EMR data available at a certain hospital. Such a process will also require selecting the most relevant covariates, e.g., from the literature[4], as computation time increases rapidly with the covariates used and might be prohibitive. However, the bootstrapping procedure we employ, which is a major factor in computation time, is easily parallelized and so could be mitigated by running the suggested pipeline on a high-performance cluster with multiple nodes. Nonetheless, our results of different magnitudes of cross-resistance between sample sources have a biological rationale, and we therefore expect them to be at least qualitatively generalizable.

Finally, our data did not include outpatient antibiotic prescriptions. This is a recurring limitation in studies of antibiotic resistance in hospitalized patients, where the patients' EMRs are not easily unified between community and hospital data sources. Nevertheless, we hope that using covariates such as patients' residency, age, comorbidities, and independence status, together with the fact that most broad spectrum and advanced-generation antibiotics are prescribed within the hospital, mitigate this issue.

To conclude, the methods we deploy in this study allow for a less biased examination of the relationships between cross-resistance and sample sources. We found similar patterns of the existence of cross-resistance in different sample sources within bacterial species; however, the magnitude of those relationships differed substantially between sample sources. Consequently, considering sample sources when assessing the likelihood of cross-resistance is crucial for methods attempting to estimate or predict antibiotic resistance. Future antibiotic prescription policies aiming to minimize collateral resistance should therefore also be differentially determined by the source of infection.

**Reporting summary**. Further information on research design is available in the Nature Portfolio Reporting Summary linked to this article.

## Data availability

The data pertain to the patients' electronic medical records. These are private and cannot be shared without approval from Meir Medical Center's IRB. Upon request, the authors and the individuals interested in accessing the data can write a formal request to the aforementioned IRB and seek its approval. The parameter estimates and credible intervals used to plot Fig. 3 are in Supplementary Data 16.

## Code availability

Sample R code for running an abn model on these data is available at https://doi.org/10.5281/zenodo.7670932[33].

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

## Acknowledgements
This study was supported by the Israel Science Foundation (ISF 1286/21).

## Author contributions
U.O. conceived the study and supervised all analysis; S.S.C. performed the analyses; M.C. provided clinical insights; all authors interpreted the results; S.S.C. and U.O. wrote the initial draft; all authors critically revised and approved the paper.

## Competing interests
The authors declare no competing interests.
