## [Peer Review File · Communications Medicine]

Reviewers' comments:

Reviewer #1 (Remarks to the Author):

Many thanks for the opportunity to review this manuscript on the use of additive Bayesian networks to study antibiotic cross-resistance across different sources. It presents an interesting proof of concept of how these data-driven approaches could help unearth patterns of cross-resistance, whilst controlling for confounders. I could not find any other publication using this method, but it appears sound provided the limitations are discussed and keeping in mind my comment below about the robustness.

Major comments

#1 Robustness of the results. While the methods are well described (pp 8 and 9), it feels very much exploratory and that the authors proceeded by trial-and-error to fine tune data processing and model hyperparameters until the model converged and the results made some sense. I appreciate that this is about applying new methods to a clinical problem, but it gives the impression that a different dataset or another team trying the same approach would have led to different results. Could the authors comment on the robustness of the analyses?

#2 Clinical relevance. The paper presents an interesting proof of concept, but could you elaborate on how you imagine that this approach will help with “future estimation of cross-resistance patterns and facilitate determination of antibiotic treatment regimens”? (in your abstract)

#3 In the discussion p14, one reason why the results differ between the present study and reference #13 may be that they relate to different cohorts of patients from different countries with different healthcare systems and antibiotic practices. It might be worth adding this.

#4 P14, when discussing the tissue that “sample size differed greatly both in bacterial species and sample sources. This translates into differing levels of statistical power across ABN analyses”. Could you not normalise the results by sample size?

Minor comments

I suggest simplifying the title: “Bayesian Network modelling...”

Abstract: it should include some values about the number of samples you analysed.

P8, assumption #6 “the variables for antibiotics given in the previous year could only be caused by age and sex and no other variable”. It could also be caused by multiple other factors including the comorbidities you have. Why not including those?

Reviewer #2 (Remarks to the Author):

In this manuscript, the authors evaluate the cross-resistance patterns of five common microorganisms *Escherichia coli*, *Staphylococcus aureus*, *Pseudomonas aeruginosa*, *Klebsiella pneumoniae*, and *Proteus mirabilis* for three different sample sources (urine, wound and aerobic blood). For that purpose, an approach previously proposed by them based on

additive Bayesian Networks (ABN).

Minor comments/questions:

1. In the literature it is common to find “Bayesian Network Modelling” however in the title the authors have used “Bayesian Networking Modelling”. Just wondering if there is any specific reason for that.
2. In the introduction, the authors briefly explain that ABN has been chosen because it allows controlling for major confounding variables and refer to their previous published manuscript. While it might be explained in the previous manuscript, what other approaches have been used to estimate cross-resistance patterns?
3. What is the absolute range of the arc strength? 0-100?
4. In the “restricting causal relations” paragraph point (8) “obesity can only be caused only by one” the word only appears twice?

Further comments:

5. The authors have restricted the model space to disallow causal paths that were not clinically plausible. The paragraph might be a bit cumbersome and would probably benefit having a Figure with the nodes and the restricted connections in red. Label each connection with a number and explain why that causal relation is not plausible.
6. In addition to these restrictions, the authors removed variables to be able to train the model within a reasonable amount of time (days rather than months). What is the feasibility of this approach if it needs to be extended to include more nodes (e.g. additional co-morbidities, antimicrobials, ...) and/or larger amount of data (e.g. from thousands to hundreds of thousands). Does it have to be retrained for every dataset? Can it be trained online, or it must be done from scratch?
7. It would be interesting to understand how these findings could be translated to clinical practice for instance how it could be used to guide antimicrobial therapy selection (e.g. single vs combination of antimicrobials) or testing.
8. Regarding reproducibility, how could other researchers try this approach? Is there any shared code or demo with dummy data to show how to configure the R package so that other researchers can verify the set-up is correct before including their own data into the model?

Overall, the main difference between this manuscript and their previously published manuscript “Revealing antibiotic cross-resistance patterns in hospitalized patients through Bayesian network modelling. 2021 Journal of Antimicrobial Chemotherapy” is the addition of more samples and the division of data by sample source. The concept behind it is interesting yet the conclusions presented are somehow vague.

Reviewer #3 (Remarks to the Author):

Firstly congratulations to the authors, i thought it an interesting piece of work. I was asked to focus on the microbiology aspects primarily, so here are a few points from me:

1. Abstract Introduction

"and especially not while considering confounding" - this sentence is missing a word

2. Main introduction

"However, such experimental conditions do not necessarily parallel bacteria living in human populations" - suggest replacing the word 'parallel' with 'mimic'

3. Main introduction - it would be worth making the point that key pathogens (including ones in the study) have intrinsic resistance to certain antibiotics; ie not all resistance is acquired

4. Materials and methods, Data

"The corresponding medical history, demographics, previous hospitalizations, and previous in-hospital antibiotic usage in the year prior to the infection..." - do you have data on community antibiotic prescriptions for these patients? would be pertinent.

5. Materials and methods, Data

"Bacterial samples were tested for antibiotic resistance for an array of antibiotics, and results of non-susceptibility and resistance were combined into a 'resistant' category." - worth mentioning if lab methodology had changed over the study period. eg changes have been made by EUCAST wrt how Pseudomonas AST is reported.

6. Table 2

couldn't see any blood culture dat for pseudomonas. also the AMC resistance for Proteus in urine and wound needs the relevant percentage documented.

7. Materials and methods, stats.

".....and a binary variable for antibiotic use in hospital in the previous year." - do authors feel a year is long enough? numerous microbiota studies show longer lasting effects of antibiotics.

School of Public Health
Porter School of Environmental and Earth Sciences

4 January 2023

Dear Editor

Thank you for giving us the opportunity to submit a revision of our manuscript “Bayesian network modelling of patterns of antibiotic cross-resistance by bacterial sample source” for consideration for publication in *Communications Medicine*. We are grateful to the reviewers for their thorough assessment of the manuscript and for their helpful suggestions for revision to improve it. Please see a point-by-point response (regular font) to the reviewers’ comments (**bold font**) below:

Reviewer #1 (Remarks to the Author):

Many thanks for the opportunity to review this manuscript on the use of additive Bayesian networks to study antibiotic cross-resistance across different sources. It presents an interesting proof of concept of how these data-driven approaches could help unearth patterns of cross-resistance, whilst controlling for confounders. I could not find any other publication using this method, but it appears sound provided the limitations are discussed and keeping in mind my comment below about the robustness.

We thank the reviewer for this positive comment.

Major comments

#1 Robustness of the results. While the methods are well described (pp 8 and 9), it feels very much exploratory and that the authors proceeded by trial-and-error to fine tune data processing and model hyperparameters until the model converged and the results made some sense. I appreciate that this is about applying new methods to a clinical problem, but it gives the impression that a different dataset or another team trying the same approach would have led to different results. Could the authors comment on the robustness of the analyses?

We are sorry we didn’t emphasize this enough in the text. However, we followed very specific standard procedures to create the ABNs, as described by Kratzer (2019, cited in the text). These procedures include an exact search procedure to identify all putative links between variables, followed by a parametric bootstrap procedure to drop those identified links which don’t have strong statistical support. The heuristic change we made pertained to limiting the number of parents, but given no model

required more than 7 parents in the end, this should not have affected the results. This is explained in the Supplementary Information Table S2. Furthermore, the full procedure and references to its different stages are given throughout the revised Methods section.

#2 Clinical relevance. The paper presents an interesting proof of concept, but could you elaborate on how you imagine that this approach will help with “future estimation of cross-resistance patterns and facilitate determination of antibiotic treatment regimens”? (in your abstract)

We thank the reviewer for this useful comment. We indeed did not stress the practical implications of our study well enough. We have now dedicated a paragraph in the Discussion to this matter:

“From a practical standpoint, our study offers two main advances. First, highlighting the consideration of source-dependent cross-resistance patterns. Studying cross-resistance *in situ* is of critical importance, since bacterial conditions in infections may substantially differ from *in vitro* conditions, leading to different adaptations [30,31]. In our study, we found that in several cases, the differences of cross-resistance magnitude between sample sources were dramatic. For instance, in *K pneumoniae*, the OR between gentamicin and ofloxacin ranged from around 6 in urine to nearly 40 in blood. In *E coli*, the same antibiotics varied in their ORs from 3 in urine to 11 in blood. These differences can distinguish between a treatment being merely unlikely to succeed, given one type of resistance, to being almost futile. Moreover, despite the complexity of the results, one general trend can be discerned: *P mirabilis* exhibited higher cross-resistance in urine than in wound samples, whereas the opposite was true for *K pneumoniae* and *P aeruginosa*. Such information could be incorporated into local clinical guidelines or decision support systems. For example, using the above estimates, if a certain hospital has high frequencies of ofloxacin resistance, an immediate implication is that gentamicin would be futile to use for bloodstream infections but may still be efficient in urinary tract infections. Second, we provide code and detailed description of the methods used to obtain our results. This will enable researchers and clinicians to analyse covariate-adjusted cross-resistance patterns on their particular datasets, and draw their specific conclusions, which may vary from ours, as discussed below.”

#3 In the discussion p14, one reason why the results differ between the present study and reference #13 may be that they relate to different cohorts of patients from different countries with different healthcare systems and antibiotic practices. It might be worth adding this.

We agree and have added it to the text:

“systematic differences in healthcare systems, antibiotic prescription practices, or cohorts of patients may also contribute to such discrepancies. For example, our data are relatively homogenous, since they originated from a single center.”

Furthermore, our revised Discussion now explains more thoroughly that differences between locations may affect the results and how this can be corrected by re-fitting our models to each dataset:

“To draw exact conclusions for a specific hospital, one would need to fit our models to the respective data, as resistance patterns may differ substantially between hospitals. This would entail retraining the models with EMR data available at a certain hospital. Such a process will also require selecting the most relevant covariates, e.g., from the literature [4], as computation time increases rapidly with the covariates used and might be prohibitive. However, the bootstrapping procedure we employ, which is a major factor in computation time, is easily parallelized and so could be mitigated by running the suggested pipeline on a high-performance cluster with multiple nodes.”

#4 P14, when discussing the tissue that “sample size differed greatly both in bacterial species and sample sources. This translates into differing levels of statistical power across ABN analyses”. Could you not normalise the results by sample size?

Smaller sample sizes reduce our ability to discover links between antibiotics when they are there, and thus render our analyses conservative (i.e., we could have found perhaps more links). When we do find links, the sample size and variance are reflected in the CIs we draw, and hence normalisation will not aid this situation. Furthermore, we note that when we do find links, most of the estimates differ significantly, indicating that we indeed have sufficient power to discover differences in those. We now explicitly refer to this issue in the limitations’ section in the expanded discussion of sample size:

“First, sample size differed greatly both in bacterial species and sample sources. This translates into differing levels of statistical power across ABN analyses. For example, we observe the most arcs in the network for *E coli* obtained from urine, potentially because the sample size was largest in that analysis. For the three other bacterial species cultured from urine, it is apparent that there are more arcs in the models connecting antibiotic resistance tests as compared to other sample sources, again potentially due to urine having the largest sample sizes. Nonetheless, while sample sizes were modest in some groups, ample data was available for others, and the study examined all bacterial samples obtained in a large hospital over a four-year period, making it quite comprehensive. Moreover, this limitation only pertains to lack of identification of further links. When we do present identified links, the influence of the sample sizes is exhibited in our uncertainty quantifications (e.g., CIs), which did successfully demonstrate differences between most sample sources.”

Minor comments

I suggest simplifying the title: “Bayesian Network modelling...”

Thanks, this was a typo. Done.

Abstract: it should include some values about the number of samples you analysed.

We updated the abstract to state the range of sample sizes available:

“Overall, the number of samples available were 3525 for *E coli*, 1125 for *K pneumoniae*, 1828 for *P aeruginosa*, 701 for *P mirabilis*, and 835 for *S aureus*.”

We then deleted the following sentence to accommodate the word restriction.

P8, assumption #6 “the variables for antibiotics given in the previous year could only be caused by age and sex and no other variable”. It could also be caused by multiple other factors including the comorbidities you have. Why not including those?

Our rationale was that only comorbidities that involve infection should directly cause prior antibiotic usage. The comorbidities we modelled do not necessarily involve infection, so it seems reasonable that they don't cause prior antibiotic usage. Age and sex were included to help reduce residual confounding, however. In any case, these restrictions should have little influence on the relationships among resistance tests, as they pertain to restriction among covariates.

Reviewer #2 (Remarks to the Author):

In this manuscript, the authors evaluate the cross-resistance patterns of five common microorganisms Escherichia coli, Staphylococcus aureus, Pseudomonas aeruginosa, Klebsiella pneumoniae, and Proteus mirabilis for three different sample sources (urine, wound and aerobic blood). For that purpose, an approach previously proposed by them based on additive Bayesian Networks (ABN).

Minor comments/questions:

1. In the literature it is common to find “Bayesian Network Modelling” however in the title the authors have used “Bayesian Networking Modelling”. Just wondering if there is any specific reason for that.

Thanks, as mentioned above this was a typo and has been fixed.

2. In the introduction, the authors briefly explain that ABN has been chosen because it allows controlling for major confounding variables and refer to their previous published manuscript. While it might be explained in the previous manuscript, what other approaches have been used to estimate cross-resistance patterns?

Mostly experimental, or simple pairwise comparisons, have been used in the past. This is mentioned in the Introduction:

“Cross-resistance of bacteria to different drugs is a commonly observed phenomenon in experimental [5–12] and clinical settings [13–17].”

Our ABN approach is the first, to our knowledge, to control for major confounding variables, as we now state in the Discussion:

“Although cross-resistance has been studied epidemiologically [13,15,20,29], such studies rarely compare cross-resistance patterns of different bacterial sample sources [13]. To the best of our knowledge, ours is the first study performing such a comparison while controlling for relevant clinical covariates.”

3. What is the absolute range of the arc strength? 0-100?

We present odds ratios, which can take any positive value.

4. In the “restricting causal relations” paragraph point (8) “obesity can only be caused only by one” the word only appears twice?

Thanks for catching this, fixed.

Further comments:

5. The authors have restricted the model space to disallow causal paths that were not clinically plausible. The paragraph might be a bit cumbersome and would probably benefit having a Figure with the nodes and the restricted connections in red. Label each connection with a number and explain why that causal relation is not plausible.

We agree that it was a bit difficult to read and tried to improve it: we have rephrased the paragraph and also present a matrix of arc restrictions in Supplementary Figure 1, with banned arcs denoted by a red box, within which is a number corresponding to the restriction described in the text.

6. In addition to these restrictions, the authors removed to variables to be able to train the model within a reasonable amount of time (days rather than months). What is the feasibility of this approach if it needs to be extended to include more nodes (e.g. additional co-morbidities, antimicrobials, ...) and/or larger amount of data (e.g. from thousands to hundreds of thousands). Does it have to be retrained for every dataset? Can it be trained online, or it must be done from scratch?

A general limitation of such machine learning/statistical models is that they need to be retrained on new data, to obtain optimal results. In particular, abn models become expensive in terms of computational time as the variables introduced into the models

(rather than the data) increase. In our case, while we did need to leave out variables in some models for them to run in a reasonable amount of time, these were variables that empirically did not relate to resistance. As a general rule, datasets can be pruned of variables that don't relate to any of the resistance tests prior to analysis, if running time is slow. This can be achieved by using prior knowledge of variables regarding how they relate to antibiotic resistance. We now added these issues to the Discussion:

“To draw exact conclusions for a specific hospital, one would need to fit our models to the respective data, as resistance patterns may differ substantially between hospitals. This would entail retraining the models with EMR data available at a certain hospital. Such a process will also require selecting the most relevant covariates, e.g., from the literature [4], as computation time increases rapidly with the covariates used and might be prohibitive. However, the bootstrapping procedure we employ, which is a major factor in computation time, is easily parallelized and so could be mitigated by running the suggested pipeline on a high-performance cluster with multiple nodes.”

7. It would be interesting to understand how these finding could be translated to clinical practice for instance how it could be used to guide antimicrobial therapy selection (e.g. single vs combination of antimicrobials) or testing.

This is a good suggestion, also mentioned by reviewer #1. We added a paragraph in the Discussion, quoted in response to that reviewer's second comment.

8. Regarding reproducibility, how could other researchers try this approach? Is there any shared code or demo with dummy data to show how to configure the R package so that other researchers can verify the set-up is correct before including their own data into the model?

We agree that this was lacking in the initial submission and thank you for the remark. The code is now fully available on github (<https://github.com/sscherny/abn-antibiotic-resistance.git>), and the detailed description in the supplementary materials should aid others in implementing it on their own data:

“Second, we provide code and detailed description of the methods used to obtain our results. This will enable researchers and clinicians to analyse covariate-adjusted cross-resistance patterns on their particular datasets, and draw their specific conclusions, which may vary from ours, as discussed below.”

Overall, the main difference between this manuscript and their previously published manuscript “Revealing antibiotic cross-resistance patterns in hospitalized patients through Bayesian network modelling. 2021 Journal of Antimicrobial Chemotherapy” is the addition of more samples and the division of data by sample source. The concept behind it is interesting yet the conclusions presented are somehow vague.

This novelty of analysis cross-resistance by sample source is indeed important as very few studies considered sample sources, and none of them controlled for patient covariates while doing so. Our study provides these results together with detailed description of the methods for achieving them and the relevant code. We have tried to better elucidate these practical implications of this study in the revised Discussion: “From a practical standpoint, our study offers two main advances. First, highlighting the consideration of source-dependent cross-resistance patterns. Studying cross-resistance *in situ* is of critical importance, since bacterial conditions in infections may substantially differ from *in vitro* conditions, leading to different adaptations [30,31]. In our study, we found that in several cases, the differences of cross-resistance magnitude between sample sources were dramatic. For instance, in *K pneumoniae*, the OR between gentamicin and ofloxacin ranged from around 6 in urine to nearly 40 in blood. In *E coli*, the same antibiotics varied in their ORs from 3 in urine to 11 in blood. These differences can distinguish between a treatment being merely unlikely to succeed, given one type of resistance, to being almost futile. Moreover, despite the complexity of the results, one general trend can be discerned: *P mirabilis* exhibited higher cross-resistance in urine than in wound samples, whereas the opposite was true for *K pneumoniae* and *P aeruginosa*. Such information could be incorporated into local clinical guidelines or decision support systems. For example, using the above estimates, if a certain hospital has high frequencies of ofloxacin resistance, an immediate implication is that gentamicin would be futile to use for bloodstream infections but may still be efficient in urinary tract infections.”

Reviewer #3 (Remarks to the Author):

Firstly congratulations to the authors, i thought it an interesting piece of work. I was asked to focus on the microbiology aspects primarily, so here are a few points from me:

Thank you for these kind words.

1.Abstract Introduction

“and especially not while considering confounding” - this sentence is missing a word

Sorry, but we are not sure it is. We double checked the sentence and it seems fine. Perhaps it got cut-off in your version:

“While cross-resistance is well-studied experimentally, it is not the case in clinical settings, and especially not while considering confounding.”

2.Main introduction

"However, such experimental conditions do not necessarily parallel bacteria living in human populations" - suggest replacing the word 'parallel' with 'mimic'

Thanks for the suggestion, done.

3. Main introduction - it would be worth making the point that key pathogens (including ones in the study) have intrinsic resistance to certain antibiotics; ie not all resistance is acquired

Good suggestion. Such resistance types are not tested in our data and hence not relevant to us. We updated the definition of cross-resistance in the introduction accordingly:

“Importantly, in this study we will only refer to cross-resistance due to mechanisms that are observed to be acquired through genetic changes, and not those inherent, for example, to a certain bacterial species.”

4. Materials and methods, Data

"The corresponding medical history, demographics, previous hospitalizations, and previous in-hospital antibiotic usage in the year prior to the infection..." - do you have data on community antibiotic prescriptions for these patients? would be pertinent.

No, unfortunately HMO and hospital data are very logistically difficult to link in Israel, but we hope to have captured such confounders via other variables. We add this as a limitation in Discussion:

“Finally, our data did not include outpatient antibiotic prescriptions. This is a recurring limitation in studies of antibiotic resistance in hospitalized patients, where the patients’ EMRs are not easily unified between community and hospital data sources. Nevertheless, we hope that using covariates such as patients’ residency, age, comorbidities, and independence status, together with the fact that most broad spectrum and advanced-generation antibiotics are prescribed within the hospital, mitigate this issue.”

5. Materials and methods, Data

"Bacterial samples were tested for antibiotic resistance for an array of antibiotics, and results of non-susceptibility and resistance were combined into a ‘resistant’ category." - worth mentioning if lab methodology had changed over the study period. eg changes have been made by EUCAST wrt how Pseudomonas AST is reported.

This is a very valid point but fortunately the methodology has not changed during the period of this study.

6. Table 2

couldn't see any blood culture dat for pseudomonas. also the AMC resistance for Proteus in urine and wound needs the relevant percentage documented.

Sample size was insufficient to model the blood cultures for *Pseudomonas* so these were not included in the analysis.

Sorry, the page break was between the counts and percentages for AMC so it was hidden in the table. We have fixed that now.

7. Materials and methods, stats.

".....and a binary variable for antibiotic use in hospital in the previous year." - do authors feel a year is long enough? numerous microbiota studies show longer lasting effects of antibiotics.

Ideally this would be entered as a continuous variable with an interaction term for people not using antibiotics previously, as our previous study has shown. This interaction is, however, not straight forward in abn modelling, so we deferred to a binary covariate. If the time period is longer, such a binary covariate may dilute the effect (i.e., people treated two years ago will be in the same category and contribute to a smaller effect size). Hence, we decided that one year is an appropriate threshold. This is now explained in the Methods:

“Although modelling previous antibiotic use is preferable as continuous with an interaction of a binary used/didn’t use term,[21] this could not be achieved under the ABN framework, so we resorted to the mentioned binarization.”

REVIEWERS' COMMENTS:

Reviewer #1 (Remarks to the Author):

Many thanks for the opportunity to review this revised manuscript. The authors have done a great job at addressing the reviewers' comments. I don't have any further input in the manuscript.

Reviewer #2 (Remarks to the Author):

Thank you very much to the authors for addressing all the comments and making their code available to other researchers. There is nothing else to add from my side.